# DNA nano-pocket for ultra-selective uranyl extraction from seawater

Yihui Yuan[1,2], Tingting Liu[1,2], Juanxiu Xiao[1], Qiuhan Yu[1], Lijuan Feng[1], Biye Niu[1], Shiwei Feng[1], Jiacheng Zhang[1] & Ning Wang [1✉]

Extraction of uranium from seawater is critical for the sustainable development of nuclear energy. However, the currently available uranium adsorbents are hampered by co-existing metal ion interference. DNAzymes exhibit high selectivity to specific metal ions, yet there is no DNA-based adsorbent for extraction of soluble minerals from seawater. Herein, the uranyl-binding DNA strand from the DNAzyme is polymerized into DNA-based uranium extraction hydrogel (DNA-UEH) that exhibits a high uranium adsorption capacity of 6.06 mg $g^{-1}$ with 18.95 times high selectivity for uranium against vanadium in natural seawater. The uranium is found to be bound by oxygen atoms from the phosphate groups and the carbonyl groups, which formed the specific nano-pocket that empowers DNA-UEH with high selectivity and high binding affinity. This study both provides an adsorbent for uranium extraction from seawater and broadens the application of DNA for being used in recovery of high-value soluble minerals from seawater.

[1] State Key Laboratory of Marine Resource Utilization in South China Sea, Hainan University, 570228 Haikou, P. R. China. [2]These authors contributed equally: Yihui Yuan, Tingting Liu. ✉email: wangn02@foxmail.com

                                                                                              

Nuclear power is one of the most mature and environment friendly energy supply[1,2]. As the fuel of the nuclear power industry, the uranium resource in the terrestrial uranium ore is estimated to only feed the nuclear power plant for less than one century, without taking into account the ever-increasing consumption of energy[3]. The seawater, with a total amounts of about 4.5 billion tons uranium reserve, is estimated to contain 1000 times more uranium than that in terrestrial uranium ore and can ensure the sustainable nuclear energy generation for thousands of years[4,5]. Thus, the highly efficient utilization of ocean uranium resource is a promising approach to meet the growth demands of uranium.

However, the extraction of uranium from the seawater is challenging, which is due to the complex ocean environment, including the high ion strength, the complicated interfering ions, the low uranium concentration (3.3 ppb), and the severe marine biofouling[6–10]. For purpose of high-efficient extracting uranium from seawater, the design of excellent uranyl binding functional group that suitable for being used in ocean environment is highly concerned. The application of amidoxime group at the early 1980s has significantly enhanced the efficiency for uranium extraction and is used to be thought as the most promising candidate for the recovery of uranium from seawater[11–14]. However, the amidoxime group is severely hazarded by the interfering ions from the seawater, especially the vanadyl, which even exhibits higher binding competitiveness to amidoxime group than uranyl[15]. Some other functional groups are also used for constructing uranium adsorbents, such as the engineered proteins[16,17], bio-inspired nanostructures[18–20], molecularly imprinted polymer[21], and the network formed by hydroxyl groups[22], and show enhanced uranium adsorption selectivity compared with the amidoxime group. These results indicate that the application of novel functional group is a promising strategy for developing novel uranium adsorbents[23–26].

The DNAzyme, which is also named DNA enzymes or catalytic DNA, is a kind of DNA that can catalyze the splitting of the single strand DNA[27]. Attributing to the special spatial structure formed by the nucleotide acid in the DNA, the DNAzyme is only activated by binding specific metal ions, including uranyl[28]. The DNAzyme is found to exhibit high selectivity to specific metal ion against the co-existing interfering metal ions[29,30]. Attributing to the highly specific binding ability to specific metal ion, the DNAzyme has been used in the detection of metal ions[31–33]. However, currently, there is no DNA-based adsorbent for the recovery of soluble minerals from seawater, and the recognition mechanism of DNAzyme to specific metal ions, including uranyl, is still unclear. Attributing to the potential applications of DNA in areas of medical science and material science, the strategy for fabrication of DNA hydrogel are highly concerned[34–36], and thus made the application of DNA in uranium extraction from natural seawater feasible. The synthesis of DNA hydrogel by using rolling circle amplification (RCA) technology is thought as an efficient approach for large-scale fabrication of DNA materials with low economic cost[37,38].

Herein, inspired by the specific recognition ability of DNAzyme to uranyl, the uranyl-binding DNA strand from the DNAzyme is used as an aptamer and polymerized into a DNA hydrogel, named as hydrogel DNA–UEH, for the extraction of uranium from seawater (Fig. 1). The DNA–UEH shows promising uranium extraction performance in natural seawater with high selectivity to uranyl against completing metal ions. The EXAFS analysis reveals the DNA bound uranium by forming coordination bonds between the uranium atom and the oxygen atoms from the phosphate groups and the carbonyl groups in the DNA molecular. The finding of this study both broadens the application of DNA and provides a kind of adsorbent for uranium extraction from seawater.

## Results

**Fabrication of DNA-based hydrogel**. To fabricate adsorbent for selective uranyl extraction, the DNA strand, originating from the DNAzyme, that showed high selectivity to uranyl ion was used to synthesize the hydrogel DNA–UEH. The sequences of the DNA strand and related primes were shown in Supplementary Table 1. The specific loop structure in the DNA strand from the DNAzyme was reported to essential for the binding of uranyl ion[39]. According to previous reports, the DNAzyme used in this study exhibited high selectivity to uranyl ions against the co-existing interfering ions and could resistant diverse environmental interference[3], which would benefit the practical application of the DNA-based hydrogel in natural seawater. The single strand from the DNAzyme that responsible for selective uranyl ion binding was firstly cyclized into circular DNA, and furthterly, the circular single DNA strand was used as template to synthesize hydrogel DNA–UEH, by rolling circle amplification (Fig. 1b)[30]. The synthesis process of the DNA hydrogel was independent from complex chemical reaction, expensive chemicals, and expensive equipment, which enhanced the potential for the practical application of the DNA hydrogel.

**Characterizations of hydrogel DNA–UEH**. The microscopic morphology of the dried DNA hydrogel existed as interacted flower-like structure composing of nanoscale petals (Fig. 2a), which was similar with the morphology of the other DNA hydrogel fabricated by RCA[34]. The newly fabricated DNA hydrogel DNA–UEH exhibited a meta-mechanical feature and existed as solid-like property in water but liquid-like property after being taken out from water (Fig. 2b). This kind of feature indicated that the DNA hydrogel maintained stable molecular network between the DNA molecular both in water, and after being taken out from the water. The removal of the environmental water would not cause damage to the adsorbent during the recovery of the adsorbent from seawater. The DNA hydrogel could be stained by the fluorochrome gel-red, suggesting that the DNA hydrogel contained double strand structure. The double strand structure was responsible for maintaining the loop structure in the DNA, which was essential for uranyl binding. Furthermore, the DNA hydrogel also showed tunable morphology by temperature treatment (Fig. 2c). After being heated to 90 °C, the shape of the DNA hydrogel could be repeatedly remolded between different shapes, indicated that the temperature treatment would influence the network of the DNA molecular in the DNA hydrogel. During the practical application of the adsorbent in ocean, the ocean wave and ocean current would damage the adsorbent and reduce the reusability of the adsorbent[14]. The hydrogel DNA–UEH showed high tensile property. After being remolded into DNA hydrogel fiber, the length of the DNA hydrogel fiber could be stretched by more than five times (Fig. 2d). The excellent mechanical properties of the DNA hydrogel were conductive to the reusability and durability of the adsorbent during practical application in ocean[14,40]. The chemical composition of hydrogel DNA–UEH was analyzed using Fourier-transform infrared spectroscopy (FT-IR). The result showed that vast phosphate groups were detected in the DNA hydrogel, which was the components of the DNA molecular (Supplementary Fig. 1).

**Determination of uranyl binding mechanism**. Although the DNA strand from DNAzyme showed high uranyl selectivity, the binding mechanism of the DNA strand to uranyl was rarely

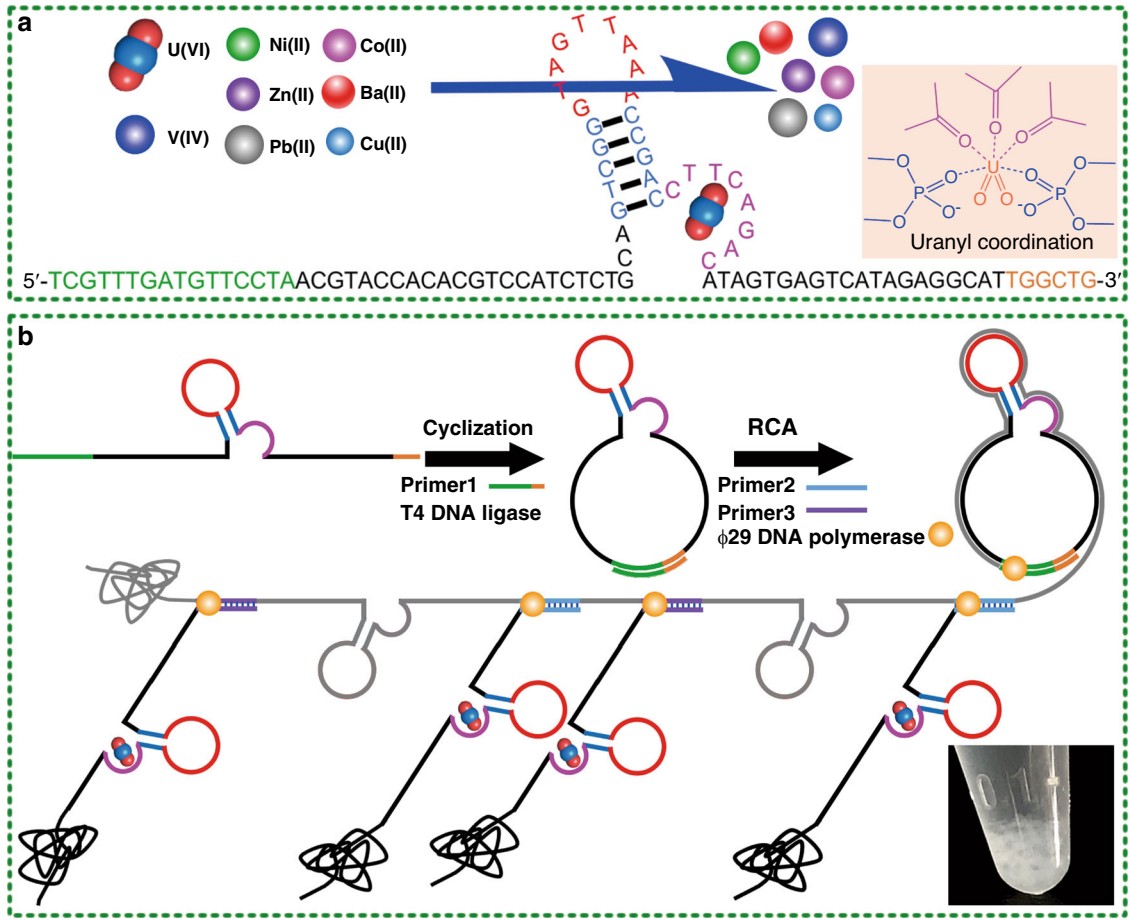

**Fig. 1 Schematic diagram for fabrication of hydrogel DNA-UEH and mechanism for uranyl ion binding by hydrogel DNA-UEH. a** Structure of uranyl-binding DNA strand from DNAzyme and mechanism for selective uranyl binding. **b** Strategy for fabrication of hydrogel DNA-UEH.

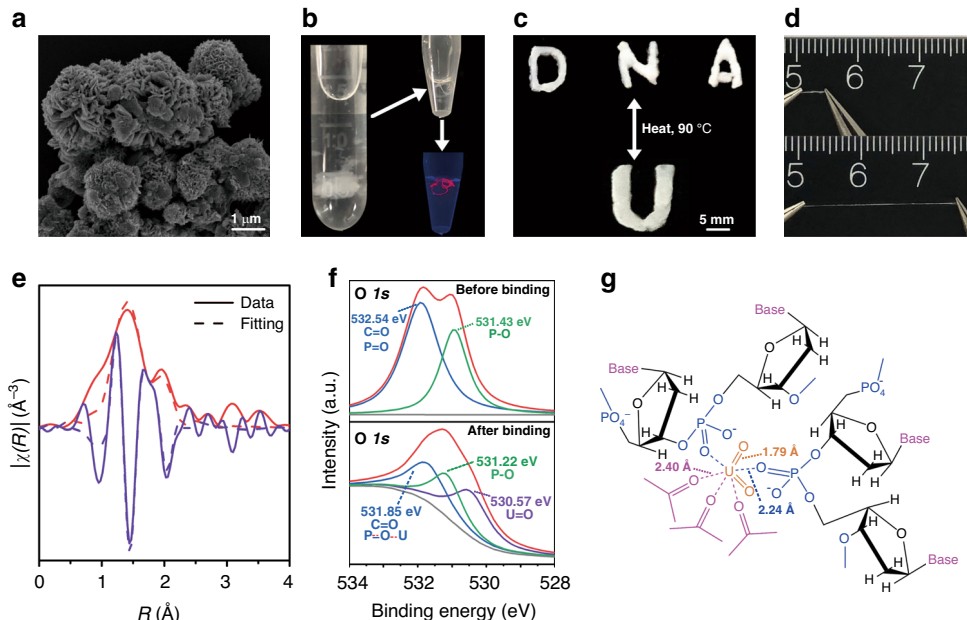

**Fig. 2 Characterization of hydrogel DNA-UEH and uranyl binding mechanism of DNA nano-pocket. a** SEM observation of the dried hydrogel DNA-UEH **b**. Morphology and fluorescence staining of the wet hydrogel DNA-UEH. **c** Remolding of the hydrogel DNA-UEH by heat treatment. **d** Tensile property of the hydrogel DNA-UEH. **e** EXAFS analysis of uranium bound by hydrogel DNA-UEH. **f** High-resolution XPS analysis of oxygen element in hydrogel DNA-UEH before and after uranyl binding. **g** Speculated coordination environment of uranyl in hydrogel DNA-UEH. The coordination bonds and the average bond lengths are shown.

analyzed. To uncover the uranium adsorption mechanism, the extended X-ray absorption fine structure spectrum (EXAFS) and X-ray photoelectron spectroscopy (XPS) analysis were engaged. The EXAFS analysis showed that three peaks at 1.79, 2.24, and 2.40 Å was detected (Fig. 2e and Supplementary Fig. 2 and Table 2). The peak at 1.79 Å corresponded to scattering paths of the axial oxygen atoms in the uranyl. The peak at 2.24 Å corresponded to the two coordination bonds of the uranium atom with the oxygen atom from the phosphate[41] and the peak at 2.4 Å corresponded to the three coordination bonds of the uranium atom with the carbonyl oxygen atom from the base of the deoxyribonucleotide[20]. These five oxygen atoms formed a penta-coordinated DNA nano-pocket structure, which is responsible for the high binding affinity and binding selectivity to uranyl. Previous functional analysis of the DNAzyme revealed that the loop structure in the single DNA strand determined the uranyl binding specificity[30] and the changes of the oxygen (P=O) in the phosphate group significantly influenced the catalytic activity of DNAzyme[27]. Thus, it is rational to speculate that the oxygen atoms from the phosphate groups at the skeleton of the DNA strand were critical for the binding of uranyl. However, the finding of this study revealed that the carbonyl oxygen atoms from the base of the deoxyribonucleotide were also essential for uranyl binding.

The XPS analysis of the hydrogel DNA–UEH before and after uranium binding showed that the uranium element was detected in the hydrogel after adsorption and the position of oxygen element was changed in the XPS spectrums (Supplementary Fig. 3). The detail high-resolution XPS analysis of the oxygen element before the binding of uranyl showed that two peaks of oxygen element were detected, representing the peak of oxygen in the phosphate group (P–O, 531.43 eV) and the peak of oxygen formed double bond with phosphorus and carbon atoms (532. 54 eV, C=O, and P=O), respectively (Fig. 2f). After the binding of the uranyl, three peaks of oxygen element were detected, including the axial oxygen of uranyl (530.57 eV), the oxygen in phosphate group (P–O, 531.22 eV), and another peak at 531.85 eV, which was the shift of peak at 532.54 eV before uranyl binding. The peak at 531.85 eV was speculated to be the overlap of the peaks for oxygen in P=O and C=O (532.54 eV) and the peak for oxygen in P–O–U and C–O–U, which was formed by the coordination of uranium atom with the oxygen atoms in P=O from phosphate group and C=O from carbonyl group (Fig. 2g). The detail XPS analysis of the uranium element showed that the uranium element maintained as it's valent in uranyl after being bound (Supplementary Fig. 4).

**Uranium adsorption capacity in uranium spiked simulated seawater**. To determine the uranium adsorption capacity of the hydrogel DNA–UEH, the uranium spiked simulated seawater was used to stimulate the salty of seawater and the species of uranyl, which existed as $[UO_2(CO_3)_3]^{4-}$ in the natural seawater. The optimal pH analysis revealed that the hydrogel DNA–UEH showed the highest uranium adsorption capacity of 13.66 mg g$^{-1}$ at pH 5.0 in 8 ppm uranium spiked simulated seawater by calculating with the dry weight of the hydrogel. At pH 8.0, which was near the pH of natural seawater, the hydrogel DNA–UEH showed a uranium adsorption capacity of 6.64 mg g$^{-1}$ (Fig. 3a). The reduction of the uranium adsorption capacity at pH 8.0 was due to the form of uranyl, which existed as $[UO_2(CO_3)_3]^{4-}$ by coordinated with carbonate group at pH 8.0. The carbonate group would complete with the DNA–UEH for uranyl and caused the drop of uranium adsorption capacity.

The uranium adsorption capacities in different concentration uranium spiked simulated seawater were used to analyze the

uranium adsorption mechanism and these uranium adsorption capacities of the hydrogel DNA–UEH fitted well with both Langmuir and Freundlich fitting models with the correction coefficients higher than 0.99 (Fig. 3b). Based on the fitting result of the Langmuir fitting model, the theoretical maximum adsorption capacity of the DNA hydrogel was calculated to be 16.15 mg g$^{-1}$, which was close to the experimental data in 16 ppm uranium spiked simulated seawater. The adsorption kinetics of the hydrogel DNA-UEH to uranium were also determined in uranium spiked simulated seawater with different uranium concentrations at pH 5.0. The result showed that the hydrogel DNA–UEH reached equilibrium adsorption capacities of 4.65, 10.17, and 13.71 mg g$^{-1}$, in uranium spiked simulated seawater with uranium concentration of 2, 4, and 8 ppm, after interacted for 24 min (Fig. 3c). After reaching the saturation adsorption, the concentrations of the uranium in the 2 ppm, 4 ppm, and 8 ppm, uranium spiked seawater were reduced to 0.83 ppm, 1.46 ppm, and 4.57 ppm, respectively (Fig. 3d). The hydrogel DNA–UEH exhibited fast adsorption speed compared with the other adsorbent used for uranium adsorption, which usually taken hours or days to reach saturation adsorption in uranium spiked solution[14,42]. The analysis of the adsorption kinetics revealed that the adsorption behavior of the hydrogel DNA–UEH fitted well with the pseudo-second fitting model at 2 ppm ($R^2$ value higher than 0.99), but the pseudo-first fitting model ($R^2$ value higher than 0.99) at 4 ppm and 8 ppm (Supplementary Fig. 5). Based on the molecular weight of the DNA strand (28,710.37 Da) and the binding mechanism that one DNA strand bound one molecular of uranyl, the theoretical chemical adsorption capacity to uranium was calculated to be 8.29 mg g$^{-1}$, which also confirmed that physical adsorption happened at high concentration uranium solution. This result indicated that the binding of uranyl by hydrogel DNA-UEH was mainly happened by chemical coordination of the DNA nano-pocket structure in low concentration uranium contained aqueous, while the physical adsorption happened in high concentration uranium contained aqueous.

The reusability of the adsorbent determined the economic cost of the adsorbent used for uranium extraction from natural seawater. The reusability of the hydrogel DNA–UEH was determined in uranium spiked simulated seawater with uranium concentration of 8 ppm at pH 5.0. The result showed the EDTA solution could elute more than 95% of the bound uranium (Supplementary Fig. 6). After been reused for five times, the hydrogel DNA–UEH still reserved 78.07% of the initial uranium binding capacity and an average 4.39% reduction of the adsorption capacity was observed after each regeneration process. The high reusability of the hydrogel DNA–UEH might be due to the meta-mechanical property of the DNA hydrogel, which maintained stable network structure both in water and after been taken out from water.

**Binding selectivity and binding affinity to uranyl**. The selectivity of the adsorbent to uranyl were critical for its' practical application in natural seawater, which contained complex interfering ions, including both electropositive metal ions and electronegative carbonate group[17,20]. To determine the selectivity of the hydrogel DNA-UEH to uranyl, the concentration of ten metal ions, including $UO_2^{2+}$, $VO^{2+}$, $Fe^{3+}$, $Co^{2+}$, $Ni^{2+}$, $Cu^{2+}$, $Zn^{2+}$, $Ba^{2+}$, $Pb^{2+}$, and $Sr^{2+}$, were improved by 100 times by adding of additional metal ions to natural seawater, and the other metal ions, including $Na^+$, $Mg^{2+}$, $Ca^{2+}$, $K^+$, maintained as their original concentration in natural seawater[43]. The result showed that, even with 100 times more interfering ions, the hydrogel DNA–UEH still exhibited high adsorption capacity to uranyl ions, but not the other metal ions (Fig. 4a). The hydrogel

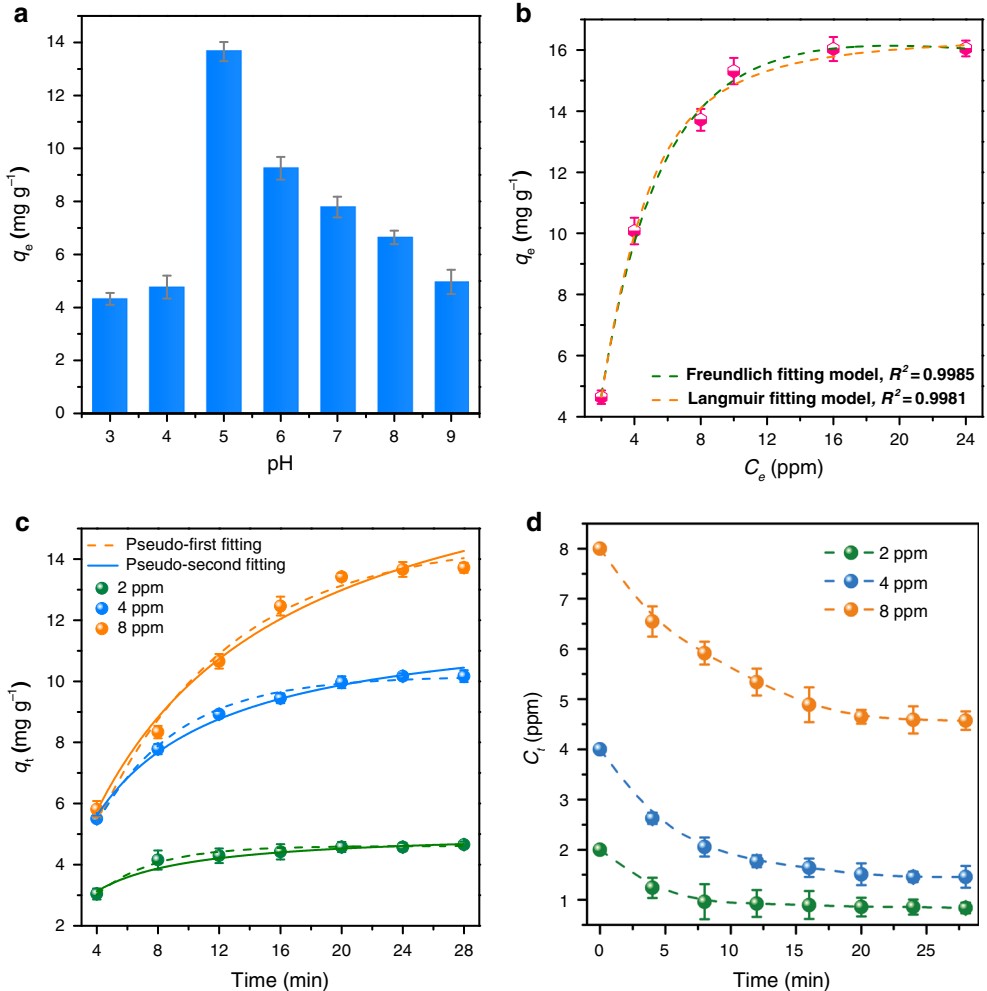

**Fig. 3 Uranium adsorption capacity in uranium spiked simulated seawater. a** Influence of pH on uranium adsorption capacity. **b** Uranium adsorption isotherms of hydrogel DNA–UEH. **c** Kinetics for uranium adsorption by hydrogel DNA–UEH in different concentration uranium spiked simulated seawater. **d** Change of uranium concentration in uranium spiked simulated seawater during the adsorption process. The error bars indicate standard deviation ($n$ = 3). (Source data provided in source data file).

DNA–UEH showed 1230.53 times higher adsorption selectivity to uranyl against vanadyl (Fig. 4b). As for the other metal ions, the hydrogel DNA–UEH showed much higher selectivity with the highest selectivity of $1.47 \times 10^5$ to $Sr^{2+}$. The influences of the high concentration metal ions from seawater, including $Na^+$, $Mg^{2+}$, $Ca^{2+}$, $K^+$, have also been analyzed at the concentrations near their concentration in natural seawater. The result showed the changes of the concentrations of these metal ions didn't cause significant change to the uranium adsorption capacity (Supplementary Fig. 7). The competition of the interfering ions to uranyl was further analyzed one by one and the result showed that the interfering ions only interfered the adsorption capacity of the hydrogel DNA–UEH to uranyl by at least $9.75 \times 10^3$ times excess (Fig. 4c). However, due to the low excess of the interfering ions in natural seawater, the interfering ions in the natural seawater was unlike to compete with the uranyl ion to the hydrogel DNA–UEH.

In natural seawater, the uranyl ion existed as stable $[UO_2(CO_3)_3]^{4-}$ by coordinating with carbonate group, which would compete with the adsorbent for the uranyl ion. Thus, for high-efficient extraction of uranium from natural seawater, the adsorbent needed to complete with the carbonate groups for uranyl. The competition between the hydrogel DNA–UEH and

the carbonate group to uranyl was tested. The result showed that a significant decrease of the fraction of bound uranyl was observed when the concentration of carbonate group was 40 times to that of uranyl, while there were only three carbonate groups in $[UO_2(CO_3)_3]^{4-}$ (Fig. 4d). Based on the competition test, the hydrogel DNA–UEH was calculated to exhibit a high binding affinity with the binding $K_d$ of 0.10 nM to the uranyl, which was highly enough for the tight binding of uranyl ion and for completing with carbonate group.

**Uranium extraction from natural seawater.** In natural seawater, the adsorption saturation was reached after adsorbing for 6 days with the final extraction capacity of 6.06 mg g$^{-1}$ to uranium element by calculating with the dry weight of the hydrogel (Fig. 5a). The final concentration of uranium reduced from 3.35 to 2.73 ppb, indicating that 18.5% of the uranium in the seawater was adsorbed by the hydrogel DNA–UEH. Corresponding with the binding selectivity in metal ions spiked natural seawater, the hydrogel DNA–UEH also showed a high selectivity to the uranyl in natural seawater. Compared with the vanadyl, which severely completed with the uranyl to the amidoxime group-based uranium adsorbent, the adsorption capacity to uranium of

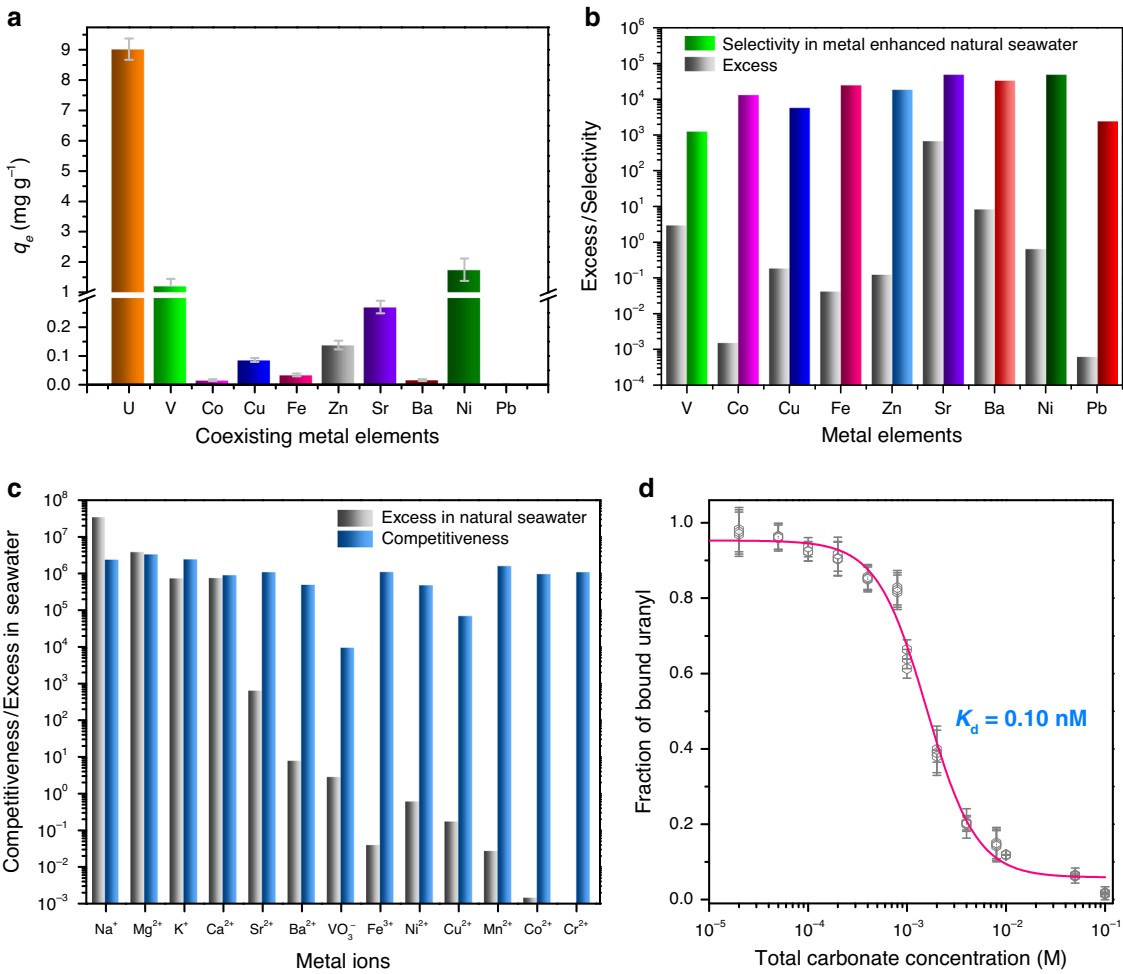

**Fig. 4 Binding selectivity and binding affinity of hydrogel DNA–UEH to uranyl. a** Adsorption capacity to 100 times concentrated coexisting interfering ions in natural seawater. **b** Binding selectivity to 100 times concentrated coexisting interfering ions. **c** Competition to individual interfering metal ion. **c** Binding affinity to uranyl against carbonate group. The error bars indicate standard deviation ($n = 3$). (Source data for A and D provided in source data file).

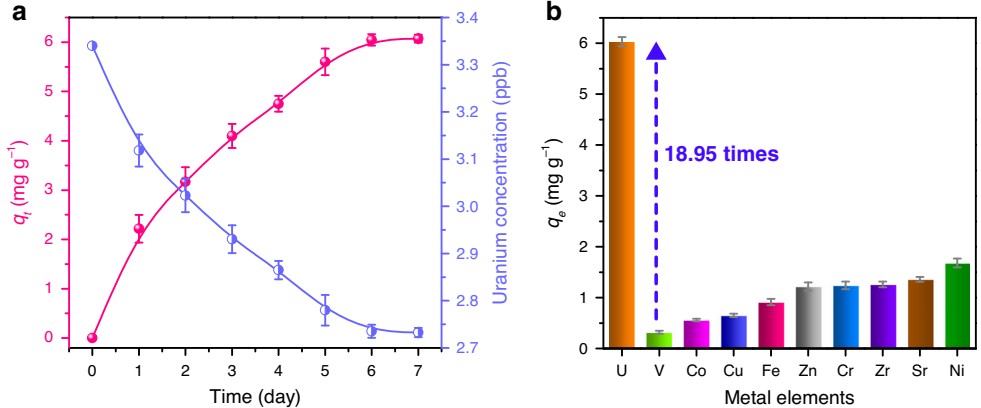

**Fig. 5 Uranium extraction from natural seawater. a** Uranium extraction kinetic of the hydrogel DNA–UEH and change of uranium concentration in natural seawater during the extraction process. **b** Extraction selectivity to metals in natural seawater. The error bars indicate standard deviation ($n = 3$). (Source data provided in source data file).

DNA–UEH was 17.95 times higher than that to vanadium (Fig. 5b), while the most widely used amidoxime group adsorbents only show 0.25–2.5 times selectivity to uranium against to vanadium. What's more, the hydrogel DNA–UEH also showed high selectivity to uranium against the other metals in natural seawater.

## Discussion

In summary, in this study, the uranyl-binding DNA strand from the DNAzyme was used in developing adsorbent for uranium extraction from seawater. The DNA strand used in this study was reported to exhibit high selectivity to uranyl. However, the binding mechanism of this DNA strand to uranyl was rarely

analyzed. Herein, the adsorption mechanism of this DNA strand to uranyl was analyzed by EXAFS and XPS analysis. The DNA strand was detected to bind the uranyl by forming five coordination bonds between the oxygen atoms in the phosphate group and the carbonyl group with the uranium atom in the uranyl, which formed a binding nanopocket that endowed the adsorbent with high binding selectivity and binding affinity to the uranyl. The high selectivity and high binding affinity efficiently avoided the influence of interfering ions and realized a high uranium extraction capacity of 6.06 mg g$^{-1}$ in natural seawater. Compared with the most art-to-the-state amidoxime group-based adsorbent for uranium extraction, the binding selectivity of the DNA hydrogel to uranium was much higher, which maintained 17.95 times higher selectivity to uranium than to vanadium in natural seawater. The hydrogel DNA–UEH showed high mechanical properties and high reusability, which would benefit the practical application of the adsorbent for uranium extraction in the ocean. The finding of this study, on one hand, developed a DNA-based adsorbent for promising uranium extraction from seawater. On the other hand, uncovered the binding mechanism of the DNA strand to uranyl and broaden the application of DNA molecular. Attributing to the specific recognition ability of the DNA strand from the DNAzyme to different high-value metal ions, this kind of DNA adsorbent could also be developed for recovering of the other high-value soluble minerals from seawater.

## Methods

**Fabrication of DNA hydrogel**. To fabricate the DNA hydrogel for uranium extraction, the single strand DNA with a sequence as shown in Supplementary Table 1 was firstly cyclized into circular DNA[34,44]. Simply, the phosphorylated DNA template and primary Primer 1 with were mixed with a mole ratio of 1:1, and then, the mixture was heated at 95 °C for 2 min, 65 °C for 2 min and gradually cooled to 60 °C at a rate of −0.5 °C every 30 s, followed by gradually cooling down to 20 °C using a PCR thermal cycler. After annealing, T4 DNA ligase and 10 × T4 DNA ligase buffer were added, and the reaction solution was incubated at 20 °C for 2 h to synthesize the circ-DNA template. To inactivate the T4 DNA ligase, the solution was firstly heated to 65 °C for 10 min. After the cyclization process, the circ-DNA templates (50 nM) were mixed with 10 U φ29 DNA polymerase, 10 μL 10 × φ29 DNA polymerase buffer, 2 μL 10 × BSA, 2 μL 800 mM NaCl, and 4 μL 25 mM dNTPs. The total volume of gelling system was filled up to 100 μL by adding of deionized water. This resultant solution was directly incubated at 30 °C for 4 h for first round of rolling circle amplification. For second round of rolling circle amplification, 1 μL Primer 2 (100 μM) and 1 μL Primer 3 (100 μM) were added into the resultant product and the mixture was incubated for additional 24 h at 30 °C. The synthesized DNA hydrogel was used for the following study.

**Characterization of the DNA hydrogel**. To determine the dry weight of the DNA hydrogel, the DNA hydrogel was firstly quick-frozen in liquid nitrogen and then lyophilized overnight by using a vacuum freeze dryer (SCIENTZ-10N). The SEM was observed by field emission scanning electron microscopy (FESEM, Hitachi-S4800). The FTIR spectra were analyzed with a PerkinElmer FTIR spectrometer. A Kratos AXIS-SUPRA spectrometer was used to analyze the X-ray photoelectron spectrum (XPS). The remold character of the DNA hydrogel was analyzed by heating the DNA hydrogel to 90 °C and then cooling to room temperature after being transferred in mold of different shapes. For EXAFS analysis, the uranyl bound DNA hydrogel was firstly dried and pressed into tablet. Furtherly, the tablet was used for EXAFS analysis at the BL14W1 substation of the Shanghai synchrotron radiation facility and detected by Lytle detector. The analysis was performed at room temperature and the Zr foli (17998 eV) was used as the reference material for energy calibration. The software Athena and Artemis were used for data processing and analysis. The single-crystal structure of triuranyl diphoshate tetrahydrate and sodium uranyl triacetate was archived from the Cambridge Structural Database (CSD) and used as starting physical model for EXAFS fitting.

**Assay of uranium adsorption capacity**. To determine the uranium adsorption capacity, the DNA hydrogel with a dry weight of 5 mg was soaked in 20 mL uranium spiked simulated seawater (438.607 mM NaCl and 2.297 mM Na$_2$CO$_3$ in deionized water) of different uranium concentration. The adsorption capacity was calculated by determining the final concentration of the uranium element with inductively coupled plasma optical emission spectrometry (ICP-OES). The optimal pH for uranium adsorption was analyzed by soaking DNA hydrogel adsorbent with a dry weight of 5 mg in 20 mL 8 ppm uranium spiked simulated seawater with pH of 3.0, 4.0, 5.0, 6.0, 7.0, 8.0, and 9.0, respectively. After being shaken for one hour at 25 °C, the concentration of the residual uranium was determined by ICP-OES.

To determine the uranium extraction capacity in natural seawater, the seawater collected from the west coast of Haikou City, Hainan Province, China, was filtered through a 0.22 μm filter to remove the marine microorganism and the suspended particles to avoid the influence of marine fouling to the uranium sequestration capacity, and then used for uranium extraction capacity assay. For each test, 100 L filtered seawater and DNA hydrogel with a dry weight of 10 mg was used and the test was performed at 25 °C. The seawater was flowing through the DNA-hydrogel at a speed of 1 L min$^{-1}$ and the concentrations of uranium in the seawater were analyzed by using inductively coupled plasma mass spectrometry (ICP-MS) at an interval of 12 h. The adsorption capacity to the other metals were also determined.

**Determination of binding selectivity and binding affinity**. To determine the selectivity of the DNA hydrogel to marine metal ions, ten metal ions, including UO$_2$$^{2+}$, VO$^{2+}$, Fe$^{3+}$, Co$^{2+}$, Ni$^{2+}$, Cu$^{2+}$, Zn$^{2+}$, Ba$^{2+}$, Pb$^{2+}$, and Sr$^{2+}$, were added into natural seawater to the concentration of 100 times of these metal ions in natural seawater[45] (Supplementary Table 3). The other main ions, including Na$^+$, Mg$^{2+}$, Ca$^{2+}$, and K$^+$, were maintained as their original concentrations in natural seawater. Subsequently, DNA hydrogel with dry weight of 10 mg was added into 500 mL of above prepared solution. After adsorbed for 1 h at 25 °C with moderate shaking, the concentration of each metals in the solution was determined by using ICP-MS. The adsorption selectivity was determined by calculating the K value of each metals, in which

$$K = \frac{U_{\text{ads}}}{U_{\text{aq}}},\qquad(1)$$

where $U_{\text{ads}}$ (mg kg$^{-1}$) is the concentration of adsorbed uranium in the adsorbent and $U_{\text{ads}}$ (mg kg$^{-1}$) is the final concentration uranium in the metal ions spiked natural seawater.

To accurately compare the competing of the other metal ions against uranium, 100 mL 500 nM uranyl ions in deionized water with pH of 6.0 mixed with different concentrations of competing metal ions (Supplementary Table 4)[17]. For each test, DNA hydrogel with dry weight of 2 mg was added into the solution and the adsorption capacity to uranium was determined by ICP-MS after being adsorbing for 1 h at 25 °C with moderate shaking. If uranium element was not detected, the concentration of the competing ions was ten-fold diluted and repeated the process until uranyl was not detected. The maximum ratio of competing ions to uranium was tested at 10$^6$ metal ions to 1 uranyl.

The binding affinity of the DNA hydrogel to uranyl ion was determined by testing the competition between the carbonate group and adsorbent to uranyl. The freshly prepared carbonate solution with a pH of 8.9 was used. All water used to prepare solutions was freshly degassed and deionized and protected against further sorption of atmospheric CO$_2$. The final solution contained 10 mM Tris-HCl (pH 8.9), DNA hydrogel with dry weight of 10 mg, 10 μM UO$_2$$^{2+}$, and carbonate group with concentration ranging from 0.02 mM to 0.1 M. The fraction of residual uranium in each solution was determined by using ICP-MS.

## Data availability
The data that support the findings of this study are available within the paper and its supplementary information files. Source data for Figs. 3a–d, 4a, 4d, 5a and Supplementary Figs. 5, 6, and 7 are supplied in the source data file. Source data are provided with this paper.

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

## Acknowledgements

This work was supported by the Hainan Science and Technology Major Project (ZDKJ2019013), the National Natural Science Foundation of China (41966009 and U1967213), and the National Key R&D program of China (2018YFE0103500).

## Author contributions

Y.Y. and T.L. designed the experiments. Y.Y., T.L., Q.Y., L.F., B.N., S.F., and J.Z. carried out the experiments. Y.Y., T.L., N.W., and J. X., contributed to the data analysis. Y.Y. and N.W. wrote and revised the manuscript.

## Competing interests

The authors declare no competing interests.
