## [Peer Review File · Nature Communications]

REVIEWER COMMENTS

Reviewer #1 (Remarks to the Author):

This manuscript reported a DNA nano-pocket for extracting UO_2^{2+} ion from seawater by using DNAzyme - hydrogel. The DNA-based sorbent was prepared by polymerizing DNA into the execution hydrogel, and it exhibits some advantages such as relatively high adsorption capacity and high specificity against vanadium. Also, the binding mechanism of "DNAzyme" was investigated. However,

DNAzyme has been well-known for its ability for specific recognition of UO_2^{2+} , and the author can easily use it to construction of DNA-based sorbent. In the manuscript, there was no sparkling thought for metal ion extraction; meanwhile, no important technical improvement for preparing absorbent was described.

Other comments

(1) metal ion-based DNAzymes were defined as a class of DNAs that can catalytically cleave the substrate strand in the presence of metal ions. Nevertheless, the author here just pay attention to the structure of DNA that can selective recognition of UO_2^{2+} , and did not use its enzymatic activity. Hence, the DNAzyme here is acutally not an enzyme just a aptamer.

(2) DNA is relatively high cost, and its use for UO_2^{2+} extraction from seawater is quite not economic.

(3) Line 57-58 "The DNAzyme is found to exhibit more than millions of times higher selectivity to specific metal ion against the co-existing interfering metal ions" is not true. Actually, the anti-interference ability for co-existing metal ions was normally < 1000 -fold.

(4) There is high content of salt in seawaters. The author should investigate the effect of high concentration of Na^+ , Mg^{2+} etc. on the extraction of UO_2^{2+} .

(5) Line 130-132 "The EXFAS analysis showed that, except for the axial oxygens of the uranyl, the uranium atom coordinated with additional six oxygen atoms with an average bond length of 2.45 Å" (Fig. 2e; Supplementary Figure 2 and Table 2). Please explain show Fig. 2e, Fig. S2 can support this conclusion. The mechanism of UO_2^{2+} was still very ambiguous.

(6) Why the authors spiked so high concentration of UO_2^{2+} to seawater? There is only about 3 ppb UO_2^{2+} to seawater, but the authors spiked 2 ppm, 4ppm and 8 ppm to it. It is nonsense for the spiking experiments using such high concentrations.

(7) The experimental condition for Binding selectivity and binding affinity to uranyl were missing. What the concentration of uranyl and coexisting ions?

(8) Line 265, the unit of temperature is missing.

(9) Why it takes 6 days for reaching the adsorption saturation during extraction from the real seawater whereas only 24 min for the spiked seawater (as stated in line 192). Why the adsorption kinetics are so different?

(10) How about the extraction efficiency? Line 268. Only 18.5% of uranium was extracted for 6 days of adsorption. The extraction efficiency seem to be quite low.

Reviewer #2 (Remarks to the Author):

The authors reported in this manuscript a DNA-based adsorbent for uranium extraction from seawater, to our knowledge, which is the first report of DNA-based uranium adsorbent. The adsorbent showed high uranium extraction capacity and high selectivity to uranyl against the other interfering metal ions. The binding mechanism of the DNAzyme to uranyl was also determined, which proved that the uranyl was bound by the nano-pocket structure formed by oxygen atoms at the skeleton of the DNAzyme. This kind of DNA-based adsorbents were also highly possible to be used for the extraction of the other high-value soluble minerals from seawater. Overall, the findings of the manuscript are very interesting and will represent an important contribution to the area of uranium extraction from seawater. This reviewer supports the publication of this work in Nature Communications after some minor revisions with the following comments addressed.

1. In Fig. 2b, the authors performed a fluorescence staining of the DNA hydrogel. What's the purpose of this experiment? As described by the authors, the DNA hydrogel was composed of double strand DNA, while the specific stem-loop structure in the single strand DNAzyme was

responsible for the uranyl binding. Thus, the complementary DNA strand was useless for uranyl binding. If this is true, the Fig. 2b and the corresponding statements in the manuscript should be deleted or removed to the supplementary file.

2. Besides the high-resolution XPS spectrum of oxygen element (Fig. 2f), the peak of the oxygen element in Supplementary Figure 3 also revealed that the form of oxygen element was changed after the binding of uranyl ion, which should also be described in the main text.

3. The authors used the simulated seawater, but not the real seawater, to simulate the form of uranyl ion in natural seawater. Why not using the uranium spiked real seawater for uranium adsorption capacity assay?

4. Although the authors described that the dry weight of the hydrogel was used in the uranium adsorption experiment at the Methods section, the information was lacking in Results section. The authors should add the statements that the adsorption capacity of the hydrogel to uranium was calculated by using the dry weight in the Results section.

5. How about the theoretical maximum adsorption capacity of the DNA hydrogel to the uranium? Now that one molecular of DNAzyme theoretically binds one molecule of uranyl ion, the theoretical maximum adsorption capacity of the DNA hydrogel can be determined by the molecular weight of the DNAzyme. Is the theoretical maximum adsorption capacity calculated by using the molecular weight close to the result that calculated by using the Langmuir fitting model.

6. Why EDTA was chosen for elution of the bound uranium? According to previous reports, the elution solution containing Na_2CO_3 and H_2O_2 was used for the elution of bound uranium from uranium adsorbents.

7. The experimental details for uranium extraction from natural seawater should be moved to the Methods section.

Reviewer #3 (Remarks to the Author):

The manuscript by Yuan et al. describes the development of novel DNAzyme-based hydrogel adsorbent for selective extraction of uranyl ions from seawater. Given the low concentration of uranium and the presence of interfering ions in the ocean, the reported results of selectivity are outstanding and mark significant progress in the field of uranium recovery from unconventional sources. Moreover, the observed fast adsorption kinetics (much better than for the state-of-the-art polyamidoxime adsorbents) makes the developed material potentially useful for industrial applications. However, despite the interesting findings, the manuscript suffers substantially from a technical point of view, and thus major revisions are necessary.

First, the mechanism of uranyl binding by DNAzyme is still unclear. The details regarding the EXAFS (wrongly abbreviated in the manuscript as EXFAS) investigations are not provided to be able to reproduce the results (sample preparation, temperature of measurements, reference material for energy calibration, data processing and analysis software are missing). In addition, a starting physical model for EXAFS fitting wasn't indicated. Did the authors try different models that are likely to provide a better fit? I am specifically concerned about bidentate coordination of uranyl by each phosphate group. Available single-crystal structures from the Cambridge Structural Database (CSD) seem to indicate preference for monodentate mode instead. Therefore, it is highly unlikely that 3 phosphate groups of DNAzyme bind uranyl with 6 oxygen atoms. Other plausible models should be probed, including, for example, monodentate phosphate binding with additional water molecules completing the equatorial coordination of uranyl. In this regard, quantum mechanical modeling/calculations coupled with EXAFS analysis could be a powerful technique to elucidate the mechanism of binding. Also, it is likely that the specific arrangement of phosphate groups, i.e. binding pocket is responsible for the high selectivity, not just the presence of phosphate groups, which typically exhibit moderate binding constants with uranyl in aqueous solution.

Lines 176-179: "The reduction of the uranium adsorption capacity at pH 8.0 was due to the feature of DNA, which exhibited negative charge at pH 8.0 by the deprotonation of the phosphate group. The negative charge of the DNA molecular repulse the approach of negative $[\text{UO}_2(\text{CO}_3)_3]^{4-}$ at higher pH environment" - this simplistic explanation could be wrong. The drop of uranium adsorption capacity at higher pH values can be better explained by thermodynamics. From the speciation diagram for U(VI) [Krestou, A.; D. Panias. "Uranium (VI) speciation diagrams in the $\text{UO}_2^{2+}/\text{CO}_3^{2-}/\text{H}_2\text{O}$ system at 25° C." European Journal of Mineral Processing & Environmental

Protection, 2004], one can see that at pH 5, uranyl predominantly exists as UO_2^{2+} and UO_2OH^+ ions, while at pH 8 and higher, very stable complex of $[\text{UO}_2(\text{CO}_3)_3]^{4-}$ appears, so the completion for uranyl binding with carbonate ions is inevitable in this pH region, causing the drop of uranium capacity exhibited by the DNAzyme material.

It is also recommended to revise this manuscript by a native English speaker as multiple grammatical errors/unclear sentences were found throughout the manuscript.

Author's Response to Reviewer 1:

General Comments:

This manuscript reported a DNA nano-pocket for extracting UO_2^{2+} ion from seawater by using DNAzyme-hydrogel. The DNA-based sorbent was prepared by polymerizing DNA into the extraction hydrogel, and it exhibits some advantages such as relatively high adsorption capacity and high specificity against vanadium. Also, the binding mechanism of "DNAzyme" was investigated. However, DNAzyme has been well-known for its ability for specific recognition of UO_2^{2+} , and the author can easily use it to construction of DNA-based sorbent. In the manuscript, there was no sparkling thought for metal ion extraction; meanwhile, no important technical improvement for preparing adsorbent was described.

Response: We thank the reviewer for taking the time to review our manuscript and the valuable comments to improve the manuscript. We also appreciate the positive comments "it exhibits some advantages such as relatively high adsorption capacity and high specificity against vanadium. Also, the binding mechanism of "DNAzyme" was investigated".

Although DNAzyme has been well-known for its ability for specific recognition of UO_2^{2+} , there is still not DNA-based adsorbent. The finding of this study would broaden the area for developing uranium adsorbent with different materials. What's more, the investigation of the uranyl binding mechanism of DNAzyme would also provide novel uranyl binding biological structure and instruct the biomimetic synthesis of chemical adsorbent by simulating the uranyl binding structure in DNAzyme. Thus, the finding

of this study would be important and interesting to researchers of the related areas and suitable for publication in Nature Communications.

Other comments:

(1) metal ion-based DNAzymes were defined as a class of DNAs that can catalytically cleave the substrate strand in the presence of metal ions. Nevertheless, the author here just pay attention to the structure of DNA that can selective recognition of UO_2^{2+} , and did not use its enzymatic activity. Hence, the DNAzyme here is acutally not an enzyme just a aptamer.

Response: We thank the reviewer for the comment. The catalytic activity of the DNAzyme is based on the recognition ability of the DNAzyme to the specific metal ions. Inspired by the specific recognition ability of the DNAzyme to UO_2^{2+} , in this study, the DNAzyme was used as aptamer and polymerized into DNA hydrogel for the extraction of uranium from seawater. We have revised the function of DNAzyme in this study in the revised manuscript at Line 67-68.

(2) DNA is relatively high cost, and its use for UO_2^{2+} extraction from seawater is quite not economic.

Response: We thank the reviewer for the comment. The fabrication strategy of DNA hydrogel used in this study is based on rolling circle amplification (RCA) technology, which can realize exponential amplification of DNA only by adding a small quantity of template DNA, DNA polymerase, and dNTPs. Thus, the RCA technology is thought as an efficient approach for large-scale fabricating DNA materials with low economic cost

(Nam, J., Jang, W. S., Kim, J., Lee, H. & Lim, C. S. Lamb wave-based molecular diagnosis using DNA hydrogel formation by rolling circle amplification (RCA) process. Biosens. Bioelectron. 142, 111496 (2019); Yata, T. et al. Efficient amplification of self-gelling polypod-like structured DNA by rolling circle amplification and enzymatic digestion. Sci. Rep. 5, 14979 (2015).) Moreover, the finding of this study also provided a novel DNA structure for highly specific capturing of uranyl from seawater, which would instruct the biomimetic synthesis of chemical adsorbent that contained similar structure in the DNAzyme and realize economic extraction of uranium from seawater by low cost chemical adsorbent.

(3) Line 57-58 “The DNAzyme is found to exhibit more than millions of times higher selectivity to specific metal ion against the co-existing interfering metal ions” is not true. Actually, the anti-interference ability for co-existing metal ions was normally < 1000-fold.

Response: We thank the reviewer for the comment. We have revised the statement about the selectivity of the DNAzyme in the revised manuscript at Line 56. According to previous reports, the DNAzyme sensor that specially activated by uranyl ion exhibited million-times selectivity against the other interfering metal ions (Liu, J. et al. A catalytic beacon sensor for uranium with parts-per-trillion sensitivity and millionfold selectivity. Proc. Natl. Acad. Sci. USA 104, 2056-2061 (2007)). What’s more, our result about the competition between uranium and the other interfering metal ions also showed that the DNAzyme showed millions of times selectivity to uranium against parts of the tested metal ions (Fig. 4c). Due to the different structure of DNA molecular,

they exhibited different selectivity to co-existing metal ions. The DNAzyme used in this study was found to form nano-pocket structure for uranyl binding, which endowed the DNAzyme with high selectivity to uranyl.

(4) There is high content of salt in seawaters. The author should investigate the effect of high concentration of Na^+ , Mg^{2+} etc. on the extraction of UO_2^{2+} .

Response: We thank the reviewer for the comment. The effect of high concentration Na^+ , Mg^{2+} , K^+ , and Ca^{2+} on the extraction of UO_2^{2+} have been investigated and added into the Supporting Information as Supplementary Figure 7. By using the concentration of Na^+ , Mg^{2+} , K^+ , and Ca^{2+} near their concentrations in real seawater, these metal ions didn't show significantly influence on the extraction of UO_2^{2+} . The competition experiment had analyzed the influence of high concentration Na^+ , Mg^{2+} , K^+ , and Ca^{2+} on the uranium extraction capacity and the result showed that these metal ions only affected the uranium adsorption capacity at extremely high concentration (Fig. 4c and Supplementary Table 4). These results revealed that the adsorbent developed in this study can resist the high salty of the seawater.

(5) Line 130-132 "The EXFAS analysis showed that, except for the axial oxygens of the uranyl, the uranium atom coordinated with additional six oxygen atoms with an average bond length of 2.45 Å" (Fig. 2e; Supplementary Figure 2 and Table 2). Please explain show Fig. 2e, Fig. S2 can support this conclusion. The mechanism of UO_2^{2+} was still very ambiguous.

Response: We thank the reviewer for the comment. According to the comment of the other reviewer, the EXAFS analysis of the coordination mechanism between uranyl and the DNAzyme was re-analyzed. The result showed that uranium in uranyl coordinated with two oxygen atoms from the phosphate group and three carbonyl oxygen atoms in the base of the deoxyribonucleotide in the DNAzyme by forming five coordination bonds, which provided a much clearer coordination mechanism in the revised manuscript at line 126-134. We have replaced the coordination mechanism in previous manuscript with the new one in the revised manuscript. The revised Fig. 2e and Fig. S2 showed the coordination environment of uranium atom in the DNAzyme and the detail parameter of the coordination environment was shown in Supplementary Table 2, which showed the number and the average length of coordination bonds. We have explained the Fig. 2e and Fig. S2 in detail in the revised manuscript at Line 126-134.

(6) Why the authors spiked so high concentration of UO_2^{2+} to seawater? There is only about 3 ppb UO_2^{2+} to seawater, but the authors spiked 2 ppm, 4ppm and 8 ppm to it. It is nonsense for the spiking experiments using such high concentrations.

Response: We thank the reviewer for the comment. The high concentration UO_2^{2+} spiked seawater was used to determine the adsorption kinetics of the adsorbent, which was widely used to analyze the uranium adsorption capacity of the other adsorbents in previous reports (Sun, Q. *et al.* Bio-inspired nano-traps for uranium extraction from seawater and recovery from nuclear waste. *Nat. Commun.* 9, 1644 (2018); Ivanov, A. S. *et al.* Siderophore-inspired chelator hijacks uranium from aqueous medium. *Nat. Commun.* 10, 819 (2019)). By using high concentration UO_2^{2+} spiked seawater or

simulated seawater, we could determine the maximum uranium adsorption capacity, which might not be determined in the low concentration natural seawater. Furthermore, the uranium adsorption capacity in high concentration UO_2^{2+} spiked seawater could be used to compare with the uranium adsorption capacities of the other adsorbents.

(7) The experimental condition for Binding selectivity and binding affinity to uranyl were missing. What the concentration of uranyl and coexisting ions?

Response: We thank the reviewer for the valuable comment. The concentration of uranyl and coexisting ions used in the experiment for binding selectivity and binding affinity assay was added into the revised Supporting Information as Supplementary Table 3 and Supplementary Table 4. More experimental condition was also described in the Method section of revised manuscript.

(8) Line 265, the unit of temperature is missing.

Response: We thank the reviewer for the comment. We have revised the manuscript and added the unit of temperature in the revised manuscript.

(9) Why it takes 6 days for reaching the adsorption saturation during extraction from the real seawater whereas only 24 min for the spiked seawater (as stated in line 192). Why the adsorption kinetics are so different?

Response: We thank the reviewer for the comment. The different adsorption kinetics was caused by the different concentration of uranyl in real seawater and in spiked seawater. In spiked seawater, the concentration of uranyl is high and the adsorption process reached equilibration faster. However, in real seawater, the concentration of

uranyl is low and the adsorbent taken much more time to reach adsorption equilibration. This phenomenon fitted well with the mechanism of the adsorption kinetic. Previous reports also revealed that the adsorbents reached equilibration faster in spiker seawater than in real seawater (Sun, Q. *et al.* Bio-inspired nano-traps for uranium extraction from seawater and recovery from nuclear waste. *Nat. Commun.* 9, 1644 (2018); Das, S. *et al.* Extracting Uranium from Seawater: Promising AF Series Adsorbents. *Ind. Eng. Chem. Res.* 55, 4110-4117 (2016); Das, S. *et al.* Extracting Uranium from Seawater: Promising AI Series Adsorbents. *Ind. Eng. Chem. Res.* 55, 4103-4109 (2016)) .

(10) How about the extraction efficiency? Line 268. Only 18.5% of uranium was extracted for 6 days of adsorption. The extraction efficiency seems to be quite low.

Response: We thank the reviewer for the comment. The uranium extraction efficiency is evaluated by the adsorption capacity of the adsorbent, but not the fraction of the uranium that extracted from seawater. The ocean contains a huge amount of seawater and the concentration of uranium even don't change after the extraction process. We used a relatively large amount of real seawater to simulate the situation in the ocean. Thus, only 18.5% of uranium was extracted. The adsorbent developed in this study showed a uranium extraction capacity of 6.06 mg g⁻¹, which is higher than most of the adsorbents use for uranium extraction seawater (Abney, C. W., Mayes, R. T., Saito, T. & Dai, S. Materials for the Recovery of Uranium from Seawater. *Chem. Rev.* 117, 13935-14013 (2017)). What's more, the adsorbent developed in this study also showed a faster adsorption speed than most of the uranium adsorbent, which even taken 2

months to reach equilibration adsorption in real seawater. Thus, the extraction efficiency of the adsorbent developed in this study is relatively high.

Author's Response to Reviewer 2:

The authors reported in this manuscript a DNA-based adsorbent for uranium extraction from seawater, to our knowledge, which is the first report of DNA-based uranium adsorbent. The adsorbent showed high uranium extraction capacity and high selectivity to uranyl against the other interfering metal ions. The binding mechanism of the DNAzyme to uranyl was also determined, which proved that the uranyl was bound by the nano-pocket structure formed by oxygen atoms at the skeleton of the DNAzyme. This kind of DNA-based adsorbents were also highly possible to be used for the extraction of the other high-value soluble minerals from seawater. Overall, the findings of the manuscript are very interesting and will represent an important contribution to the area of uranium extraction from seawater. This reviewer supports the publication of this work in Nature Communications after some minor revisions with the following comments addressed.

Response: We thank the reviewer for taking time to review our manuscript and the valuable comments in improving the manuscript. We also appreciate the reviewer for the high comments to our work.

1. *In Fig. 2b, the authors performed a fluorescence staining of the DNA hydrogel. What's the purpose of this experiment? As described by the authors, the DNA hydrogel was composed of double strand DNA, while the specific stem-loop structure in the single strand DNAzyme was responsible for the uranyl binding. Thus, the complementary DNA strand was useless for uranyl binding. If this is true, the Fig. 2b*

and the corresponding statements in the manuscript should be deleted or removed to the supplementary file.

Response: We thank the reviewer for the valuable comment. The purpose of the experiment was to prove the double strand structure of the DNA molecular in the DNA hydrogel. Although the stem-loop structure in the single strand is responsible for the uranyl binding, the region of the complementary DNA strand is necessary for maintaining the structure of the DNA molecular. Thus, the Fig. 2b and the corresponding statements is useful for the manuscript. We have added more statements to describe the necessity of the experiment in the revised manuscript at Line 106-107.

2. Besides the high-resolution XPS spectrum of oxygen element (Fig. 2f), the peak of the oxygen element in Supplementary Figure 3 also revealed that the form of oxygen element was changed after the binding of uranyl ion, which should also be described in the main text.

Response: We thank the reviewer for the valuable comment. The change of oxygen element in Supplementary Figure 3 has been described in the revised manuscript at Line 143.

3. The authors used the simulated seawater, but not the real seawater, to simulate the form of uranyl ion in natural seawater. Why not using the uranium spiked real seawater for uranium adsorption capacity assay?

Response: We thank the reviewer for the comment. The uranyl in real seawater is mainly existed as $[\text{UO}_2(\text{CO}_3)_3]^{4-}$ by coordinating with the carbonate group in seawater.

The simulated seawater contained sufficient carbonate group to coordinate with the uranyl to form $[\text{UO}_2(\text{CO}_3)_3]^{4-}$, which can simulate the species of uranyl in real seawater. However, the concentration of carbonate group in real seawater is low and can't simulate the species of uranyl after adding high concentration uranyl. Thus, the simulated seawater is used for uranium adsorption capacity assay.

4. Although the authors described that the dry weight of the hydrogel was used in the uranium adsorption experiment at the Methods section, the information was lacking in Results section. The authors should add the statements that the adsorption capacity of the hydrogel to uranium was calculated by using the dry weight in the Results section.

Response: We thank the reviewer for the comment. The statements that the adsorption capacity of the hydrogel to uranium was calculated by using the dry weight were added in the revised Result section.

5. How about the theoretical maximum adsorption capacity of the DNA hydrogel to the uranium? Now that one molecular of DNAzyme theoretically binds one molecule of uranyl ion, the theoretical maximum adsorption capacity of the DNA hydrogel can be determined by the molecular weight of the DNAzyme. Is the theoretical maximum adsorption capacity calculated by using the molecular weight close to the result that calculated by using the Langmuir fitting model.

Response: We thank the reviewer for the comment. The molecular weight of the single strand DNAzyme is 28710.37 Da and the molecular weight of the uranium atom is 238 Da. According to the theory that one molecular DNAzyme binds one molecular of

uranyl, the theoretical adsorption capacity is calculated to be 8.29 mg g⁻¹. As described in the manuscript, the adsorption behavior of the hydrogel DNA-UEH fitted well with the pseudo-second fitting model at 2 ppm (R^2 value higher than 0.99), but the pseudo-first fitting model (R^2 value higher than 0.99) at 4 ppm and 8 ppm (Supplementary Figure 5). The result suggested that the high theoretical maximum adsorption capacity archived from the Langmuir fitting model was caused by physical adsorption, but not targeting chemical binding. We have added the result in the revised manuscript at Line 197-201.

6. Why EDTA was chosen for elution of the bound uranium? According to previous reports, the elution solution containing Na₂CO₃ and H₂O₂ was used for the elution of bound uranium from uranium adsorbents.

Response: We thank the reviewer for the comment. The EDTA is a metal-chelator that can tightly bind uranyl ion and the other metal ions. Thus, the EDTA was chosen to elute all the metal ions adsorbed by the adsorbent. Several of previous reports also chosen EDTA as elution solution to elute the bound uranyl ion (*Nat Chem* 2014, 6, 236-241; *Angew. Chem. Int. Ed.* 2019, 58, 11785-11790). Due to the potential damage of H₂O₂ to the DNA-based adsorbent, the elution solution containing Na₂CO₃ and H₂O₂ was not chosen.

7.The experimental details for uranium extraction from natural seawater should be moved to the Methods section.

Response: We thank the reviewer for the comment. The experimental details for uranium extraction from natural seawater has been moved to the Methods section in the revised manuscript.

Author's Response to Reviewer 3:

The manuscript by Yuan et al. describes the development of novel DNAzyme-based hydrogel adsorbent for selective extraction of uranyl ions from seawater. Given the low concentration of uranium and the presence of interfering ions in the ocean, the reported results of selectivity are outstanding and mark significant progress in the field of uranium recovery from unconventional sources. Moreover, the observed fast adsorption kinetics (much better than for the state-of-the-art polyamidoxime adsorbents) makes the developed material potentially useful for industrial applications. However, despite the interesting findings, the manuscript suffers substantially from a technical point of view, and thus major revisions are necessary.

Response: We thank the reviewer for taking the time to review our manuscript and the valuable comments to improve the manuscript. We also appreciate the reviewer for the positive comments to our work. The technical comments of the reviewer have been responded point-by-point as follows.

First, the mechanism of uranyl binding by DNAzyme is still unclear. The details regarding the EXAFS (wrongly abbreviated in the manuscript as EXFAS) investigations are not provided to be able to reproduce the results (sample preparation, temperature of measurements, reference material for energy calibration, data processing and analysis software are missing). In addition, a starting physical model for EXAFS fitting wasn't indicated. Did the authors try different models that are likely to provide a better fit? I am specifically concerned about bidentate coordination of uranyl by each phosphate group. Available single-crystal structures from the

Cambridge Structural Database (CSD) seem to indicate preference for monodentate mode instead. Therefore, it is highly unlikely that 3 phosphate groups of DNAzyme bind uranyl with 6 oxygen atoms. Other plausible models should be probed, including, for example, monodentate phosphate binding with additional water molecules completing the equatorial coordination of uranyl. In this regard, quantum mechanical modeling/calculations coupled with EXAFS analysis could be a powerful technique to elucidate the mechanism of binding. Also, it is likely that the specific arrangement of phosphate groups, i.e. binding pocket is responsible for the high selectivity, not just the presence of phosphate groups, which typically exhibit moderate binding constants with uranyl in aqueous solution.

Response: We thank the reviewer for the comments. We have corrected the writing of the EXAFS in the revised manuscript. The details for the EXAFS analysis, including the sample preparation, the temperature of measurements, the reference material for energy calibration, and the data processing and analysis software have been provided in the revised manuscript at Line 327-335. The starting physical model has also been provided. In detail, the uranyl bound DNA hydrogel was firstly dried and pressed into tablet. Furtherly, the tablet was used for EXAFS analysis at the BL14W1 substation of the Shanghai synchrotron radiation facility and detected by Lytle detector. The analysis was performed at room temperature and the Zr foli (17998 eV) was used as the reference material for energy calibration. The software Athena and Artemis were used for data processing and analysis. The single-crystal structure of triuranyl diphosphate tetrahydrate and sodium uranyl triacetate archived from the Cambridge Structural

Database (CSD) was used as starting physical model for EXAFS fitting.

According to the suggestion of the reviewer, we have re-analyzed the EXAFS data by different models, including the monodentate phosphate binding model. The result showed the monodentate model with phosphate fitted better with the data. In the revised manuscript, we have changed the previous model with the monodentate model and a new mechanism for uranyl binding was provided. The new binding mechanism showed that uranium was coordinated with two oxygen atoms from two phosphate group. What's more, the uranium also coordinated with three carbonyl oxygen atoms from the base of the deoxyribonucleotide in the DNAzyme. However, due to the lack of the crystal structure of the DNAzyme used in this study, it was impossible to carry out the calculation of the binding mechanism. As the suggestion of the reviewer, we have also explained the mechanism for the high selectivity, which was caused by the specific spatial structure of the DNAzyme that formed nano-pocket structure for uranyl binding, in the revised manuscript.

*Lines 176-179: "The reduction of the uranium adsorption capacity at pH 8.0 was due to the feature of DNA, which exhibited negative charge at pH 8.0 by the deprotonation of the phosphate group. The negative charge of the DNA molecular repulse the approach of negative $[UO_2(CO_3)_3]^{4-}$ at higher pH environment" - this simplistic explanation could be wrong. The drop of uranium adsorption capacity at higher pH values can be better explained by thermodynamics. From the speciation diagram for U(VI) [Krestou, A.; D. Panias. "Uranium (VI) speciation diagrams in the $UO_2^{2+}/CO_3^{2-}/H_2O$ system at 25° C." *European Journal of Mineral Processing &**

Environmental Protection, 2004], one can see that at pH 5, uranyl predominantly exists as UO_2^{2+} and UO_2OH^+ ions, while at pH 8 and higher, very stable complex of $[UO_2(CO_3)_3]^{4-}$ appears, so the competition for uranyl binding with carbonate ions is inevitable in this pH region, causing the drop of uranium capacity exhibited by the DNAzyme material.

Response: We thank the reviewer for the valuable comments. We agree with the reviewer about the reason that caused the drop of the uranium adsorption capacity at pH 8.0 than at pH 5.0. The thermodynamics theory has been used to explain the reason for reducing of the uranium adsorption capacity in the revised manuscript at Line 173-176. The reduction of the uranium adsorption capacity at pH 8.0 was due to the form of uranyl, which existed as $[UO_2(CO_3)_3]^{4-}$ by coordinated with carbonate group. The carbonate group would compete with the DNA-UEH for uranyl binding and caused the drop of uranium adsorption capacity.

It is also recommended to revise this manuscript by a native English speaker as multiple grammatical errors/unclear sentences were found throughout the manuscript.

Response: We thank the reviewer for the comment. We have invited a native English speaker to revise the language of the manuscript. We have also carefully revised the grammatical errors and unclear sentences in the revised manuscript.

REVIEWERS' COMMENTS

Reviewer #1 (Remarks to the Author):

In the revised manuscript, although the authors solved some of the question raised by the reviewers, the main concerns, i.e., lack of novelty and the high cost using DNA as the sorbent, were not addressed well.

(1) DNA-based aptamer was found as early as in the 1990, and it has been widely used for extraction (aptamer-based-sorbents for sample treatment-A review, *Anal. Bioanal. Chem.*, 407, 681-698), the author just extended the application of aptamer, i.e., extraction metal ions from seawater. In the response, the authors stated "the finding of this study also provided a novel DNA structure for highly specific capturing of uranyl from seawater" Actually, the structure for uranyl DNAzyme has been studied well by Liu et al. (*Proceedings of the National Academy of Sciences USA*, 2007, 104, 2056-2061), hence the author did not provided "a novel DNA structure".

(2) Author the authors used RCA for DNA amplification, the cost of synthesizing DNA was also relatively high. The materials that includes template DNA, DNA polymerase, and four types of dNTP are quite expensive for preparation of DNA in huge-scale. The RCAs for DNA amplification was mainly used for sensing application that need just a tiny amount of DNA (might be several nonogram) just as the references in the responses to question 2. The use of DNA-aptamer for extraction of metal ions from waste water or sea water which need huge-scale of DNA is not a good research topic in term of the high cost of DNA.

(3) metal ion-based DNAzymes were defined as a class of DNAs that can catalytically cleave the substrate strand in the presence of metal ions. Nevertheless, the author here just pay attention to the structure of DNA that can selective recognition of UO_2^{2+} , and did not use its enzymatic activity. Hence, the DNAzyme here is acutally not an enzyme just a aptamer. However, in the whole manuscript, the authors still used "DNAzyme".

Reviewer #2 (Remarks to the Author):

The authors have satisfactorily addressed all the comments from the reviewer and this manuscript can now be accepted as is.

Reviewer #3 (Remarks to the Author):

The authors revised the manuscript and it is now suitable for publication.

Author's Response to Reviewer 1:

In the revised manuscript, although the authors solved some of the question raised by the reviewers, the main concerns, i.e., lack of novelty and the high cost using DNA as the sorbent, were not addressed well.

Response: We thank the reviewer for taking time to review our manuscript again and the valuable comments to improve the manuscript. We have revised the manuscript accordingly as detailed in the response below. The corresponding changes have been highlighted in the revised main text. The concerns raised by the reviewer have been responded point-by-point detailed as follows.

(1) DNA-based aptamer was found as early as in the 1990, and it has been widely used for extraction (aptamer-based-sorbents for sample treatment-A review, *Anal. Bioanal. Chem.*, 407, 681-698), the author just extended the application of aptamer, i.e., extraction metal ions from seawater. In the response, the authors stated “the finding of this study also provided a novel DNA structure for highly specific capturing of uranyl from seawater” Actually, the structure for uranyl DNAzyme has been studied well by Liu et al.(*Proceedings of the National Academy of Sciences USA*, 2007, 104, 2056-2061), hence the author did not provided “a novel DNA structure”.

Response: We appreciate the reviewer for the valuable comments. According to the publication (aptamer-based-sorbents for sample treatment-A review, *Anal. Bioanal. Chem.*, 407, 681-698) mentioned by the reviewer, the DNA aptamer were mainly used for extracting of small organic molecules, proteins, cells, and only few studies about the application of DNA aptamer in separation of divalent metal ions was reported.

Furthermore, there is still no study about the extraction of uranium from natural environment by using DNA aptamer. The low selectivity is one of the most important factors that hazard the uranium extraction capacity of currently available adsorbent. The application of novel high selective ligand to construct adsorbent is a promising strategy to enhance the uranium extraction capacity of the adsorbent in seawater. Thus, the application of ultra-high selective DNA aptamer in this study is novelty and important for the area of uranium extraction from seawater.

The specific uranyl binding structure in DNAzyme is developed by Liu et al., which is the most important basis of this study. Based on this basis, we have made several important progresses, including the fabrication of high-performance DNA hydrogel for uranium extraction from seawater and the resolution of the coordination mechanism between uranyl ion and the DNA molecular. The sentence “the finding of this study also provided a novel DNA structure for highly specific capturing of uranyl from seawater, which would instruct the biomimetic synthesis of chemical adsorbent that contained similar structure in the DNAzyme” in previous response had not been expressed clearly. There is no such kind of statement in the main text. We intended to state that the finding of this study figure out the detail coordination structure between the uranyl ion and the DNA molecular, which would instruct the biomimetic synthesis of chemical adsorbent.

(2) Author the authors used RCA for DNA amplification, the cost of synthesizing DNA was also relatively high. The materials that includes template DNA, DNA polymerase, and four types of dNTP are quite expensive for preparation of DNA in huge-scale. The

RCA for DNA amplification was mainly used for sensing application that need just a tiny amount of DNA (might be several nanogram) just as the references in the responses to question 2. The use of DNA-aptamer for extraction of metal ions from waste water or sea water which need huge-scale of DNA is not a good research topic in term of the high cost of DNA.

Response: We appreciate the reviewer for the valuable comments. This study not only focused on the development of hydrogel for uranium extraction from seawater, but also determined the uranyl coordination mechanism of the DNA molecular. One of the principal values of this study is to prove the feasibility of extracting uranium from natural seawater by using DNA based adsorbent. Based on the finding of this study, the biomimetic synthesis of chemical adsorbent would greatly decrease the economic cost for uranium extraction from seawater. The amplification of DNA by using RCA is just one of the strategies for preparation of large amount of DNA. In addition, there are several other technologies for economic synthesis of DNA (Large-scale de novo DNA synthesis: technologies and applications, Nature Methods, 2014, doi:10.1038/nmeth.2918), which even can realize the extremely low cost synthesis of DNA at the cost of \$1 per 10^3 – 10^5 bp DNA and would greatly reduce the cost for huge-scale synthesis of DNA. Furthermore, along with advance of DNA synthesis technology, the economic cost for preparation of huge amount of DNA will be further greatly reduced. Just as the economic cost for the amplification of DNA by using PCR technology has been greatly reduced in the past few years.

(3) metal ion-based DNAs were defined as a class of DNAs that can catalytically cleave the substrate strand in the presence of metal ions. Nevertheless, the author here just pay attention to the structure of DNA that can selective recognition of UO_2^{2+} , and did not use its enzymatic activity. Hence, the DNAzyme here is acutally not an enzyme just an aptamer. However, in the whole manuscript, the authors still used "DNAzyme".

Response: We appreciate the reviewer for the valuable comments. We agree with the reviewer that the DNAzyme was only used as aptamer in this study. We have revised the manuscript about the use of "DNAzyme" and change the "DNAzyme" to "uranyl-binding DNA strand from DNAzyme" in corresponding sites of the revised manuscript.